# Study on Surfactant–Polymer Flooding after Polymer Flooding in High-Permeability Heterogeneous Offshore Oilfields: A Case Study of Bohai S Oilfield

**DOI:** 10.3390/polym16142004

**Published:** 2024-07-12

**Authors:** Yingxian Liu, Lizhen Ge, Kuiqian Ma, Xiaoming Chen, Zhiqiang Zhu, Jirui Hou

**Affiliations:** 1CNOOC China Limited Tianjin Branch, Tianjin 300459, Chinagelzh2@cnooc.com.cn (L.G.); 18810390092@163.com (K.M.);; 2Research Institute of Unconventional Petroleum Science and Technology, China University of Petroleum (Beijing), Beijing 102249, China

**Keywords:** heavy oil reservoir, offshore, polymer flooding, surfactant–polymer flooding, enhanced oil recovery

## Abstract

Polymer flooding is an effective development technology to enhance oil recovery, and it has been widely used all over the world. However, after long-term polymer flooding, a large number of oilfields have experienced a sharp decline in reservoir development efficiency. High water cut wells, serious dispersion of residual oil distribution and complex reservoir conditions all bring great challenges to enhance oil recovery. In this study, the method of enhancing oil recovery after polymer flooding was studied by taking the S Oilfield as an example. A surfactant–polymer system suitable for high-permeability heterogeneous oilfields was developed, comprising biogenic surfactants and polymers. Microscopic displacement experiments were conducted using cast thin sections from the S Oilfield, and nuclear magnetic resonance was employed for core displacement experiments. Numerical simulation experiments were also conducted on the S Oilfield. The results show that the enhanced oil recovery mechanism of the surfactant–polymer system is to adjust the flow direction, expand the swept volume, emulsify crude oil and reduce interfacial tension. Surfactant–polymer flooding proves to be effective in improving recovery efficiency, significantly reducing the time of flooding and further enhancing the strong swept area. The nuclear magnetic resonance results indicate a high amplitude of passive utilization of residual oil during the surfactant–polymer flooding stage, highlighting the enormous potential for an increased recovery ratio. Surfactant–polymer flooding emerges as a more suitable technique to enhance oil recovery in the post polymer-flooding stage in high-permeability heterogeneous oilfields.

## 1. Introduction

In recent years, with the declining discovery of new petroleum reserves, enhancing oil recovery (EOR) technologies have become pivotal for meeting future energy demands [1]. Well-known chemical EOR techniques include polymer flooding, surfactant flooding, and alkaline flooding [2,3]. Among these, polymer flooding has been employed in the petroleum industry for over 40 years [4], demonstrating the potential to increase the recovery efficiency of geological reserves by 5–30% [5]. China, being the largest petroleum producer in chemical EOR projects, primarily relies on polymer flooding [6,7]. The first and second largest oilfields implementing polymer flooding in China are Daqing Oilfield and Shengli Oilfield. The oil production of these two oilfields mainly depends on polymer flooding [8,9,10,11]. Moreover, there is a growing emphasis on polymer flooding in heavy oil reservoirs and offshore oilfields.

While polymer flooding has been industrialized worldwide, its potential to enhance oil recovery in practical field applications is considerably limited [12]. The mechanism of polymer flooding involves increasing the viscosity of the displacing fluid, reducing the flowability contrast between the displacing fluid and crude oil, thereby mitigating the fingering effect [13]. The flowability of the displacing phase equals or is lower than that of the oil phase [14,15]. During the process of polymer flooding displacing crude oil, interactions such as electrostatic and London dispersion forces occur between polymer molecules and the rock surface [16]. Under conditions of high reservoir salinity and elevated formation temperatures, polymer viscosity loss occurs. These factors result in the final viscosity of the injected agent in the reservoir being lower than the targeted viscosity, thereby diminishing the effectiveness and efficiency of polymer flooding [17]. Due to the severe heterogeneity of the reservoir, immature well patterns, and high temperatures with elevated salinity, approximately half of the petroleum reserves remain in the reservoir after polymer flooding [18]. The current challenges with polymer flooding include limited improvement in sweep efficiency and low oil displacement efficiency [19]. Enhancing the sweep efficiency coefficient (also known as consistency control) is a prerequisite for increasing oil recovery after polymer flooding [20]. The continuous expansion of polymer injection in oilfields, along with the gradual increase in polymer solution concentration, has led to issues such as excessive injection pressure and injection difficulties in injection wells [21].

After polymer flooding refers to the late stage of polymer flooding with a high water cut, and Residual polymer makes it difficult to effectively displace crude oil during this stage. In order to enhance oil recovery in reservoirs after polymer flooding, a series of new technologies are put forward to improve oil recovery, such as profile control, water plugging, surfactant–polymer flooding (SP flooding), ternary compound flooding (ASP flooding) and foam composite flooding. A large number of studies have shown that profile control can effectively adjust the water absorption profile, and water plugging can effectively block the high-permeability layer, thus expanding the swept volume and improving the displacement effect. SP flooding, ASP flooding and foam composite flooding have different advantages and can improve the swept efficiency [22]. The investigated literature has studied the exploitation technology and difficulties of offshore oilfields. Shuang Liang, in order to improve the development effect of offshore oil fields at the high water cutting stage, studied the technology of nitrogen foam flooding and polymer microsphere flooding [23]. Lizhen Ge clarified the mechanism of weak gel flooding and remaining oil distribution in the LD Oilfield [24]. There are relatively few studies on the application of SP flooding technology in offshore oilfields. In order to study EOR methods after polymer flooding in offshore oilfields and explore new technologies to enhance oil recovery, the S Oilfield was taken as an example to carry out corresponding theoretical and experimental research in this paper.

The S Oilfield is situated offshore of the Bohai Bay Liaohe Basin, characterized by a cover-type semi-horst structure developed on the pre-Tertiary limestone basement, trending in a north-east direction with a simple structural configuration [25]. The sedimentary type is front-edge deposition of a river delta, predominantly featuring subaqueous distributary channels and estuarine dam sand bodies. The S Oilfield belongs to heavy oil reservoirs with significant differences in underground crude oil properties. The reservoir temperature of the S Oilfield is 65 °C, the reservoir pressure is 13.5 MPa, and the viscosity of crude oil is 24~452 mPa · s. The water content of the S Oilfield is 69.2%, and the recovery degree is 26.8%. Currently, the oilfield is in the late stage of polymer injection, characterized by a high water cut, and the remaining oil is primarily located in areas where polymer flooding has not penetrated or has low sweep efficiency, such as near faults [26]. The distribution of remaining oil is more scattered and complex, posing significant challenges to recovering these reserves in polymer-flooded blocks.

The S Oilfield faces several difficulties: Firstly, due to the complex offshore reservoir conditions with multiple vertical control layers in a single well, and limitations imposed by offshore platforms, implementing measures such as layer adjustments and well pattern improvements to address the oil–water issue is challenging. Secondly, the offshore platform lacks freshwater resources required for polymer injection, relying instead on formation water and seawater with elevated salinity. Thirdly, the S Oilfield has large well spacing, severe reservoir heterogeneity, and the polymer tends to channel into high-permeability layers, complicating the recovery of remaining oil in wells [23,24,27]. If a standalone polymer flooding method continues to be employed, it will adversely impact the development effectiveness of polymer flooding [28].

Currently, the enhanced oil recovery (EOR) technology after polymer flooding in offshore oilfields requires the integration of adjustment, plugging, and displacement techniques to expand the sweep volume, thereby enhancing the oil displacement efficiency. ASP compound flooding encounters challenges such as alkali scaling and corrosion [29]. Due to environmental and platform constraints in offshore oilfields, the difficulty in handling produced fluids is greater than in onshore fields. Therefore, under current technological conditions, ASP compound flooding is not suitable for use in offshore oilfields [30]. Due to limited space on offshore platforms, it is preferable to use the equipment and injection processes already employed in polymer flooding after polymer flooding. Considering the onsite conditions and production circumstances in offshore oilfields, the S Oilfield recommends the use of an SP system combining polymers and surfactants for EOR after polymer flooding. The surfactant–polymer system used in this study is composed of a biological surfactant and hydrophobically associated polymer. This kind of surfactant–polymer system, with a low concentration of surfactants and good ability to reduce interfacial tension, can be effectively applied to high-permeability heavy oil reservoirs in the Bohai Sea.

## 2. Methods and Materials

### 2.1. Surfactant–Polymer System

In the surfactant–polymer system, the surfactant system primarily comprises lipopeptide biosurfactants as the main agent, with sodium petroleum sulfonate as an auxiliary agent for compounding. In the biosurfactant system, the ratio of lipopeptide to the auxiliary agent is 1:1, with a total mass concentration of 0.34%. The polymer selected is a hydrophobically associated polymer. The polymer concentration is set at 3000 mg/L. Operating at 60 °C with an underground viscosity of 50 cP, the interfacial tension between the surfactant–polymer system and S crude oil can reach 1.1 × 10^−3^ mN/m.

The biosurfactant used is Surfactin, produced by Maclean, with a purity of 99%. It has a molecular weight of 1036.34, a boiling point of 1268.3 ± 65.0 °C and a density of 1.037 ± 0.06 g/cm^3^. It is derived from sodium subtilis lipopeptide. The sodium petroleum sulfonate is also produced by Maclean, with a purity of 50%. The hydrophobically associated polymer is AP-P4. AP-P4 has good viscosity enhancement, shear resistance, temperature resistance up to 70 °C, salt resistance up to 10,000 mg/L, calcium and magnesium ion resistance up to 500 mg/L and good mobility control. The formation water has a mineralization level of 8000 mg/L, with calcium and magnesium ion concentrations of 240 mg/L.

The configuration method for the surfactant–polymer system is as follows:(1)Weigh 500 mL of formation water and set the stirrer to a speed of 300 rpm for stirring. Weigh 6000 mg of the hydrophobically associated polymer and slowly add it to deionized water to avoid the occurrence of fisheye phenomena due to rapid addition. If fisheye occurs, reconfigure. After the polymer is dissolved, reduce the stirring speed to 80 to avoid damaging the polymer chain structure due to excessive speed.(2)Weigh 500 mL of formation water, and set the stirrer to a speed of 180 rpm for stirring. Weigh 0.34 g of the main agent and 0.64 g of the auxiliary agent, add them separately to the formation water, and stir. After uniform stirring, add the biosurfactant system to the stirred polymer. Stir at low speed until the biosurfactant system and polymer are completely dissolved. The surfactant–polymer system configuration is complete.

### 2.2. Design and Fabrication of Micro-Physical Model

(1)Similarity Criteria

Optimizing the design of indoor experimental conditions through similarity criteria aims to make the flow patterns in the model similar to the actual reservoir flow patterns. The primary similarity indicator is chosen to be motion similarity, matching the similarity of reservoir conditions with those of the S Oilfield. The specific results are shown in Appendix A. The formula is as follows:$$Re=\frac{\rho VL}{\mu }$$
where

*ρ* is the fluid density, measured in kg/m^3^; *V* is the fluid velocity, measured in m/s; *L* is the characteristic length, measured in meters; and *μ* is the fluid viscosity, measured in Pa·s.

(2)Model Design

The design and fabrication of the model involve using the actual core structure from the S Oilfield to closely simulate the geometric structure of the core. Initially, the real core is sliced, and these core slices are used to create cast thin sections. The actual structure of the cast thin sections is then outlined to represent the particle structure, and this is transformed into a CAD model design. Based on the model design, an accurate chemical etching method is employed on glass to create the core’s pore structure, and the model is subsequently sealed. The resulting micro-scale visualization model is proportionally scaled to match the actual core dimensions, as depicted in Figure 1.

(3)Experimental Instruments and Procedures

The instruments used for microscopic displacement experiments include a dual-cylinder pump, intermediate container, six-way valve, pressure gauge, microscopic model holder, microscopic model imaging system, and connecting pipelines, as shown in Figure 2. The working pressure of the air pump ranges from 0 to 20 MPa, with a working flow rate of 0 to 15 mL/min. The volume of the intermediate container is 50 mL, with a working pressure of 0 to 40 MPa. The working pressure of the microscopic model holder is 0 to 5 MPa, as shown in Figure 3. The microscopic model imaging system comprises a microscope (Leica DM450, Leica Biosystems, Shanghai, China) and a computer. The dimethyldichlorosilane used in the experiment is produced by Aladdin, with a purity greater than 98.5%. The crude oil used is S crude oil, with an underground oil viscosity ranging from 24 to 452 cP; a viscosity of 86.5 cP is selected. The surface oil density is 0.971 t/m³.

The experimental condition of microscopic model displacement is temperature 60 °C and pressure 15 Mpa, which is consistent with the actual reservoir temperature and pressure. Furthermore, the pore shape of the microscopic model is consistent with that of the actual core, which means the experimental results are similar to those of the actual reservoir.

The experimental steps are as follows:

① Age the model using dimethyldichlorosilane, allowing it to stand for 12 h to change the wettability of the model surface to oil wet.

② Utilize a saturated model with the S Oilfield crude oil and let it stand for 24 h.

③ Use formation water from the S Oilfield to inject water into the model until the produced fluid’s water cut reaches 98%. Record the experimental process.

④ Inject polymer into the model at a rate of 0.05 mL/min until the produced fluid’s water cut reaches 70%. Record the experimental process.

⑤ Inject the SP system into the model at a rate of 0.05 mL/min until the produced fluid’s water cut reaches 98%. Record the experimental process.

### 2.3. Core Flooding Experiment

The experimental equipment used for core flooding is the Newmaze Analysis High-Temperature High-Pressure Displacement Nuclear Magnetic Resonance Testing System (MesoMR-HTHP, Suzhou Newmag Analytical Instrument Corporation, Suzhou, China). This system includes online nuclear magnetic resonance instruments, nuclear magnetic resonance analysis devices, a high-temperature high-pressure core holder, a high-temperature high-pressure displacement device, and connecting pipelines, as shown in Appendix A. The pressure range of the high-temperature high-pressure core holder is 0 to 20 MPa, and the temperature range is 0 to 80 °C. The high-temperature high-pressure displacement device is of the MR-HTHP type, with a pressure range of 0 to 20 MPa and a temperature range of 0 to 80 °C. The formation water used in the experiment is simulated formation water containing Mn^2+^, prepared from manganese chloride and deionized water. The T_2_ signals were converted into oil saturation in pores through mathematical processing, as shown in Appendix A.

The core oil displacement experiment steps are as follows:(1)Begin by washing the core and measuring its length, diameter, permeability, and weight. Subsequently, evacuate the core, saturate it with simulated formation water, and scan the nuclear magnetic resonance T_2_ spectrum upon completion.(2)Employ simulated formation water containing Mn^2+^ for constant-rate displacement of the core, thoroughly replacing the simulated formation water to eliminate water signals within the core.(3)Utilize crude oil at a constant rate of 0.5 mL/min for core displacement, continuing until the outlet of the core reaches 100% oil saturation. Establish the initial oil saturation and measure the nuclear magnetic resonance T_2_ spectrum after oil saturation.(4)Employ simulated formation water at a constant flow rate of 0.5 mL/min for water flooding, continuing until the outlet water cut reaches 70%. Monitor nuclear magnetic resonance signals in real-time during the experiment and record liquid and oil production at different time intervals.(5)Conduct polymer flooding at a constant rate of 0.3 mL/min. Halt polymer injection when 0.4 pore volumes (PVs) have been injected, followed by continuous water flooding at 0.3 mL/min until the outlet water cut reaches 98%. Pause simulated formation water injection, continuously monitor nuclear magnetic resonance signals, and record liquid and oil production at different time intervals.(6)Employ an SP system for constant-rate displacement at 0.3 mL/min. Halt the injection when 0.3 PV of the SP system has been introduced, followed by continuous water flooding at 0.3 mL/min until the outlet water cut reaches 98%. Continuously monitor nuclear magnetic resonance signals during the experiment and record liquid and oil production at different time intervals.

## 3. Results and Discussion

### 3.1. Mechanism of Enhanced oil Recovery by Surfactant–Polymer Flooding

#### 3.1.1. Improve Displacement Efficiency

Surface tension experiments were carried out by varying the mass fractions of surfactants in contact with S crude oil, and the experimental results are presented in Figure 4. It is evident from the graph that the interfacial tension was significantly reduced by the surfactant solution, reaching values below 10^−3^ mN/m. Specifically, within the range of 0.30 wt% to 0.36 wt% of surfactant mass fraction, the interfacial tension varied between 0.25 and 0.8 × 10^−3^ mN/m.

In the SP system, a robust emulsification effect on crude oil is demonstrated by surfactants. Surfactant molecules can form large-sized micelles, partially encapsulating the crude oil in the aqueous solution of surfactants, leading to the creation of oil-in-water emulsions, as illustrated in Figure 5a. Emulsions are less prone to adsorption in the pores, thereby improving the flowability of crude oil in the pores, reducing residual oil, and enhancing oil recovery. The mass fractions of surfactant solutions achieving ultra-low interfacial tension were 0.30 wt%, 0.32 wt%, 0.34 wt%, and 0.36 wt%. The emulsion droplet size distribution and average size were measured. As the concentration of surfactant increased, the average droplet sizes were 712.4 ± 200 nm, 1562.0 ± 200 nm, 955.4 ± 100 nm, and 1222.1 ± 100 nm, respectively.

The position and width of the peak in the emulsion droplet size distribution curve indicate the emulsification efficiency. A peak closer to the larger size and a wider width correspond to poorer emulsification, whereas a smaller droplet size and a narrower distribution indicate higher emulsion stability. From the peak widths of different surfactants, it can be observed that the emulsion formed with 0.34 wt% surfactant had the narrowest peak, indicating a higher level of droplet size uniformity. The emulsions formed with 0.32 wt% and 0.36 wt% surfactant follow, while the emulsion with 0.30 wt% surfactant exhibits lower droplet size uniformity, as shown in Figure 5b.

#### 3.1.2. Extend the Swept Volume

The micro-scale displacement experiments in the S model revealed that water flooding was significantly influenced by the viscosity contrast between oil and water. Upon entering the channels, water induced severe fingering effects, resulting in poor displacement efficiency for the oil on both sides. Water flooding created high-permeability flow channels, and subsequent water injection progressed along these high-permeability channels. This led to a substantial amount of remaining oil in the unreached areas, as depicted in Figure 6a.

Following the subsequent injection of polymers, the viscosity in the polymer allows for the control of the flow velocity in high-permeability zones, effectively managing the mobility and expanding the sweep volume. However, even after polymer flooding, significant residual oil remained in both the swept and unswept regions, with the distribution of residual oil in the swept region being more complex than after water flooding. Subsequent water flooding post polymer flooding encountered difficulties in displacing the remaining oil in the swept region, as illustrated in Figure 6b.

The SP system served to increase the viscosity of the aqueous phase, reducing its mobility and effectively improving the oil–water mobility ratio, thereby mitigating lateral fingering phenomena. Residual oil that could not be displaced during the polymer flooding stage experienced an increase in frictional forces with the SP system due to the reduced mobility ratio. The distribution of residual oil post SP flooding as the SP system began to flow is depicted in Figure 7. There existed mutual adsorption between the SP system and the pore walls, leading to significant frictional forces during displacement. This reduction in the flow velocity of the SP system in the pores diminished the velocity difference between high-permeability and low-permeability channels. The injection of the SP system adjusted the water imbibition profile, expanding the sweep volume and passively mobilizing the residual oil on both sides of the model, as illustrated in Figure 6c.

The micro-scale displacement experiments in the S model revealed that, on the one hand, the polymer in the SP system effectively reduced the oil–water mobility ratio, thereby expanding the sweep volume coefficient. On the other hand, the surfactant in the SP system lowered the interfacial tension between oil and water, emulsifying the crude oil and consequently enhancing the overall oil recovery.

### 3.2. Potential to Improve Oil Displacement Efficiency

After conducting ultimate water flooding experiments using three viscosity grades of crude oil, namely 45.7 cP, 86.5 cP, and 291.1 cP, the total injected volume for ultimate water flooding of the three viscosity grades of crude oil was around 230 pore volumes (PVs). The ultimate water flooding efficiencies for the three viscosity grades were 75.23%, 73.22%, and 69.87%, with effective enhancements of 25.23%, 28.99%, and 33.67%, respectively, as illustrated in Figure 8.

Selecting crude oil with a viscosity of 86.5 cP, ultimate polymer flooding and ultimate SP flooding experiments were conducted. In the case of the S Oilfield, the ultimate polymer flooding increased the recovery rate from 73.22% to 85.24%, showcasing a substantial potential for improved oil recovery. The total injected volume for ultimate polymer flooding was reduced from 230 PV in ultimate water flooding to 45 PV, marking a reduction of 185 PV and a reduction rate of 80.43%. For ultimate SP flooding, the total injected volume decreased from 230 PV to 3 PV, indicating a reduction of 227 PV and a reduction rate of 98.7%. The ultimate SP flooding achieved the maximum oil recovery efficiency more rapidly compared to polymer flooding.

Comparing the results of ultimate flooding experiments in the S Oilfield, it is evident that during the high water cut period in water flooding, both polymer and SP flooding can effectively enhance oil recovery. The EOR measure begins to be implemented at about 80% of water content. Polymer flooding reaches the displacement limit at 0.88 PV and 13.1 PV. SP flooding starts at 1.08 PV and reaches the displacement limit at 12.7 PV. The difference in ultimate oil recovery efficiency between polymer flooding and SP flooding is negligible. However, SP flooding significantly reduces the total injected PV from 45 PV to 3 PV, as depicted in Figure 9.

After polymer flooding in the S Oilfield, the potential for improvement in ultimate oil recovery efficiency ranges from 10% to 14%. Measures such as SP flooding can effectively enhance the ultimate oil recovery efficiency, reducing the total injected PV required to approach the ultimate efficiency. This can significantly lower the time and cost associated with oil recovery.

### 3.3. Potential to Expand Swept Volume

Using numerical simulation methods and focusing on the Bohai S Oilfield, this study delved into the development potential of the oilfield, with a particular emphasis on key indicators such as target recovery rate and maximum sweep coefficient. Detailed data on the geology, reservoir properties, reservoir characteristics, and development methods of the S Oilfield were utilized for a comprehensive analysis. The parameters in Table 1 were designed to establish a high-permeability, heterogeneous conceptual model of the S Oilfield. The purpose of this model was to gain profound insights into the development prospects of the S Oilfield. Through predictions of key indicators like target recovery rate and sweep coefficient, it aimed to provide theoretical support for the oilfield’s development.

The foundational geological parameters are as follows: a five-spot pattern well arrangement, with well spacing of 175 m × 150 m and reservoir thickness of 48 m, vertically divided into three distinct permeability layers (from top to bottom, each with a thickness of 8 m and permeabilities of 4813 mD, 2987 mD, and 2459 mD, respectively). The model volume is set at 600 × 10^4^ m^3^.

Applying polymer flooding followed by SP flooding to the conceptual model of the S Oilfield under a five-spot pattern well arrangement allows for the determination of the target recovery rate and maximum sweep coefficient (as shown in Figure 10). The injection rate is set at the actual field well group daily injection rate of 6340 m^3^. Utilizing data provided by the S Oilfield, a standard relative permeability curve is obtained (as shown in Figure 11). Reservoir engineering methods are then employed to calculate the maximum sweep coefficient and target recovery rate. This comprehensive assessment aims to evaluate the development potential of the S Oilfield.

(1)Evaluation of Ultimate Oil Recovery Efficiency Potential

Assuming a 100-year production period, an initial water flooding simulation was conducted. When the water cut reached 70%, a polymer with a concentration of 0.3 mg/L was injected at 0.4 PV. After the polymer injection, a second water flooding was initiated. When the water cut reached 98% in the second water flooding, an SP system with a concentration of 0.34 mg/L was injected at 0.3 PV. Subsequently, three consecutive water flooding cycles were simulated, as illustrated in Figure 12.

The graph indicates that at the end of the first water flooding, the water cut was 70%, and the recovery rate was 11.34%. During the first water flooding, the pressure initially increased rapidly to 48.04 MPa, and as crude oil was gradually produced, the pressure rapidly decreased. When the cumulative injection volume reached 0.096 PV (end of the first water flooding), the pressure finally decreased to 27.49 MPa. The pressure decrease suggests that after a period of water flooding, the reservoir has formed high-permeability channels and water injection tends to flow through these high-permeability layers, influencing the crude oil in the high-permeability layers.

When the water cut reaches 70%, polymer injection is initiated. After polymer injection, a rapid pressure increase to 66.23 MPa is observed, maintaining a brief stability, while the water cut increases to 83.83%. This phenomenon is attributed to the significant viscosity of the polymer, leading to retention in the high-permeability layer due to chemical adsorption upon entering the porous reservoir. This retention increases flow resistance, resulting in a rapid increase in injection pressure. As a large amount of water is introduced into the reservoir through the first water flooding, the water cut does not immediately decrease after polymer injection. In reality, the water cut only decreases rapidly when the polymer passes through the advantageous channels formed by the sealed water flooding. It is noteworthy that at a cumulative injection volume of 0.157 PV, the pressure begins to slowly decrease, followed by a rapid decrease. Simultaneously, the recovery rate rapidly increases, and the water cut decreases rapidly to 60.14%. This indicates that the polymer has successfully temporarily plugged the major channels in the high-permeability layer (4813 mD), redirecting subsequent fluids to other layers, thus expanding the sweep and enhancing oil recovery, leading to a rapid increase in recovery rate. When the cumulative injection volume reaches 0.28 PV, the water cut begins to increase rapidly, and the increase in recovery rate slows down. At a cumulative injection volume of 0.494 PV (end of polymer injection), the water cut increases to 92.62%, the recovery rate increases to 32.71%, and the pressure decreases to 45.32 MPa. In the second water flooding stage, the rate of pressure reduction, as well as the rates of increase in water cut and recovery rate, slows down. Ultimately, at the end of the second water flooding, the cumulative injection volume reaches 1.39 PV, the pressure is 40.64 MPa, the water cut is 98.01%, and the recovery rate is 37.74%.

When the water cut reaches 98%, an SP system is injected at 0.3 PV, and it is observed that the pressure rapidly rises to 49.3 MPa, followed by a sharp decline. Simultaneously, there is a rapid decrease in water cut and a rapid increase in recovery rate. After the completion of the SP flooding, three consecutive water flooding cycles are conducted, and it is found that the recovery rate continues to increase rapidly, indicating the successful entry of polymer-carrying surfactant into more pore volumes (see Figure 12). This allows the washing effect of the surfactant to be effectively utilized, generating a synergistic effect. In this process, the main mechanisms include the temporary plugging effect of the polymer in the SP system and the alteration of rock wettability by the surfactant. By changing the wettability of the rock surface from oil wet to water wet, the third water flooding can still ensure a rapid increase in recovery rate. It is noteworthy that when the cumulative injection volume reaches 2.23 PV, the rate of increase in recovery rate begins to slow down, and the water cut gradually rises. When the water cut reaches 98% again, the recovery rate is 62.87%. Furthermore, during the third water flooding, the surfactant’s ability to alter rock wettability remains effective, achieving continuous displacement of various oil layers and increasing crude oil production with a relatively long effective period. When the cumulative injection volume reaches 119.5 PV, the final recovery rate can reach 79.59%.

Through comparison, it is observed that at the extreme water cut of 98%, the recovery rate of polymer flooding followed by subsequent water flooding increases by 26.4%, while the recovery rate of SP system flooding followed by subsequent water flooding increases by 25.13%. Although the degrees of recovery rate improvement are similar between the two methods, the latter has a lower injection pressure, faster effectiveness, less injection volume, and shorter exploitation period, making it cost effective for field applications.

(2)Maximum Swept Efficiency

Based on the observations from Figure 13, after polymer flooding and six months of subsequent water flooding, there is still a significant amount of remaining oil in the swept area, indicated by the red and light-yellow regions. Calculations using reservoir engineering methods show a moderate sweep efficiency of 61.04% and a strong sweep efficiency of 37.11%. This indicates that after polymer flooding, the reservoir is still primarily characterized by moderate sweep efficiency.

In contrast, after the injection of the SP system and six months of three consecutive water flooding cycles, it is evident that the injection of the SP system has further enhanced its effectiveness in recovering remaining oil. The calculations reveal a moderate sweep efficiency of 34.66% and a strong sweep efficiency of 65.34% for the SP system. Compared to polymer flooding, the injection of the SP system has increased the strong sweep efficiency of the reservoir by 28.23%. This injection significantly expands the extent of reservoir sweeping, slightly reduces the dead oil zone areas on both sides, and greatly reduces the oil saturation. In summary, this indicates that the reservoir in the S Oilfield has the potential for long-term development, and the adoption of appropriate enhanced oil recovery techniques is likely to effectively promote the efficient development and utilization of the S Oilfield after polymer flooding.

The swept area under water flooding up to the extreme water cut is primarily characterized by moderate sweep efficiency (constituting 50% to 90%). Polymer flooding up to the limit helps increase the extent of sweeping, with some of the moderate swept areas transforming into strong swept areas (constituting 15% to 30%). The application of the SP system further enhances the strong swept regions.

### 3.4. Potential to Enhanced Oil Recovery

During online nuclear magnetic resonance (NMR) core flooding experiments for oil displacement, a significant reduction in the signal amplitude of the T_2_ spectrum was observed as the water displacement pore volume (PV) increased, as shown in Figure 14a. Passive utilization was predominantly observed throughout the water flooding phase, with higher amplitudes. Notably, the medium and large pores exhibited the highest degrees of mobilization, accounting for 47.66% and 50.11%, respectively, as depicted in Figure 15a. Conversely, mobilization in small pores was relatively limited. In the water flooding stage, the medium and small pores contributed the most to oil displacement, with respective contribution rates of 59.30% and 20.78%. The contribution rate of large pores was 18.34%, while that of micro-pores was comparatively lower, resulting in a stage recovery rate of 43.25%, as illustrated in Figure 16. The distribution of remaining oil after water flooding was highest in the medium pores at 53.81%, followed by small pores at 23.40%. Large pores contained a smaller portion of saturated oil at 15.83%, and micro-pores exhibited the least saturation at 6.97%, as shown in Figure 17a. Following the conclusion of water flooding, the remaining oil was predominantly distributed in the medium and small pores, with medium pores being the primary reservoir, followed by small pores. Utilization in micro-pores remained relatively low, and the proportion of remaining oil was relatively unchanged, as indicated in Figure 14b.

With an increase in polymer flooding pore volume (PV), there was a significant reduction in the signal amplitude of the T_2_ spectrum, as illustrated in Figure 14a. Throughout the polymer flooding phase, there was an overall higher passive utilization amplitude, with the highest degree of mobilization observed in micro-pores at 34.03%. The mobilization levels in other pore types were approximately around 25%, as depicted in Figure 15b. In the polymer flooding stage, the most substantial oil displacement occurred in small and medium pores, contributing rates of 25.15% and 47.81%, respectively. In comparison to water flooding, the utilization in micro-pores showed an increased contribution rate, resulting in a stage recovery rate of 25.81% and an overall recovery rate of 69.06%, as shown in Figure 16. The distribution of remaining oil after polymer flooding revealed that medium pores had the highest proportion at 28.17%, followed by small pores at 14.41%. Large pores exhibited a comparatively lower proportion of remaining oil at 7.89%, and micro-pores had the least remaining oil at only 6.28%, as depicted in Figure 17b. Following the conclusion of polymer flooding, the remaining oil was predominantly distributed in medium and small pores, with a certain degree of mobilization observed in micro-pores and a decline in remaining oil in large pores, as indicated in Figure 14b.

With an increase in SP flooding pore volume (PV), there was a significant reduction in the signal amplitude of the T_2_ spectrum, as depicted in Figure 14a. Throughout the SP flooding phase, there was an overall higher passive utilization amplitude, with the highest degree of mobilization observed in micro-pores and small pores at 20.33% and 15.74%, respectively, as shown in Figure 15c. In the SP flooding stage, the contribution rates of micro-pores and small pores showed a noticeable increase, reaching 10.20% and 26.54%, respectively, resulting in a stage recovery rate of 17.39% and an overall recovery rate of 86.45%, as illustrated in Figure 16. The distribution of remaining oil after SP flooding indicated that medium pores had the highest proportion at 15.83%, followed by small pores at 7.92%. Large pores exhibited a comparatively lower proportion of remaining oil at 6.17%, and micro-pores had the least remaining oil at only 3.09%, as shown in Figure 17c. Following the conclusion of SP flooding, the remaining oil was predominantly distributed in medium pores with a relatively high proportion at 8.91%, while the proportions of remaining oil in small and micro-pores decreased, as indicated in Figure 14b.

## 4. Conclusions

Through the aforementioned experiments in high-permeability heterogeneous offshore oilfields, it was observed that, compared to polymer flooding, surfactant–polymer flooding not only expanded the swept volume but also enhanced oil recovery on the basis of polymer flooding. This is specifically manifested in the following aspects:Microscopic displacement experiments revealed that, in the surfactant–polymer system, polymers effectively reduced the oil–water mobility ratio and increased the volumetric sweep efficiency coefficient, and concurrently, surfactants in the surfactant–polymer system reduced the interfacial tension of oil and water, emulsifying crude oil, thereby enhancing crude oil recovery.Following polymer flooding, the potential for incremental oil recovery efficiency ranged from 10% to 14%. SP flooding effectively increases the incremental oil recovery efficiency, reduces the total injected PV required to approach the incremental efficiency, and significantly lowers the cost of oil recovery time.Driving to the ultimate limit in polymer flooding contributes to an increase in the extent of sweep, with a portion of the medium sweep region transforming into a strong sweep region (constituting 15% to 30%). SP flooding can further elevate the extent of the strong sweep region.In situ nuclear magnetic resonance core flooding experiments verified the tremendous potential of SP flooding in enhancing oil recovery. Taking the S Oilfield as an example, SP flooding can increase recovery by 13.83% on the basis of polymer flooding.

## Figures and Tables

**Figure 1 polymers-16-02004-f001:**
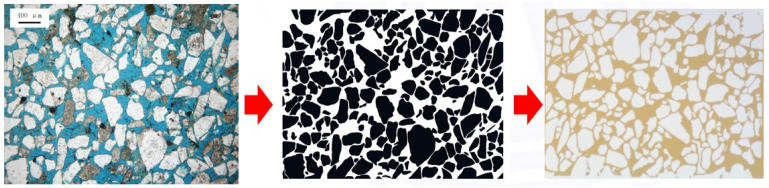
The process of making micro models for the S Oilfield.

**Figure 2 polymers-16-02004-f002:**
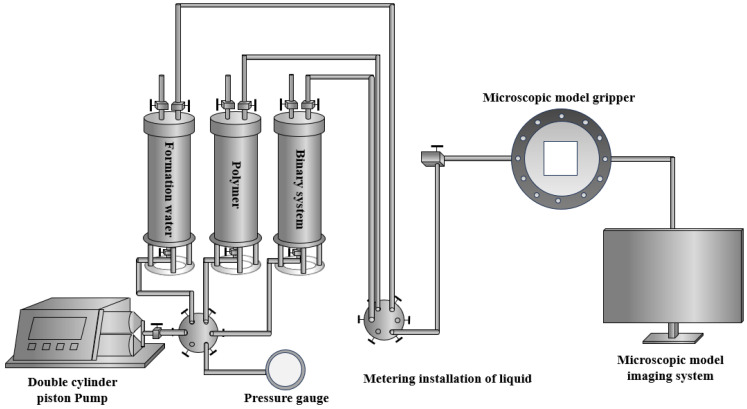
Microscopic model displacement experiment flowchart.

**Figure 3 polymers-16-02004-f003:**
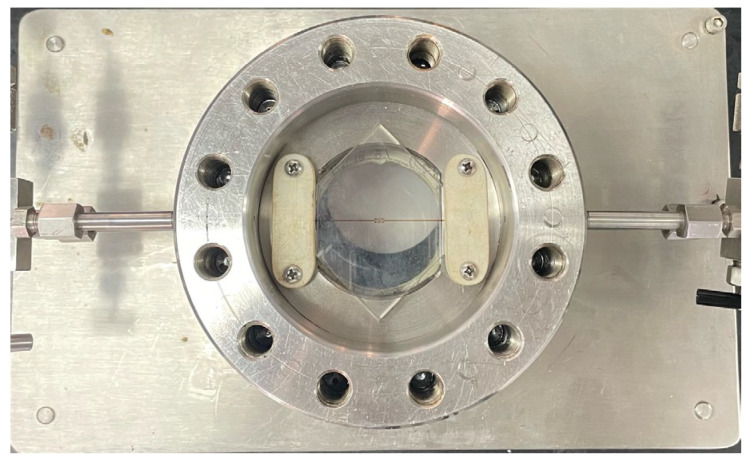
Microscopic model gripper.

**Figure 4 polymers-16-02004-f004:**
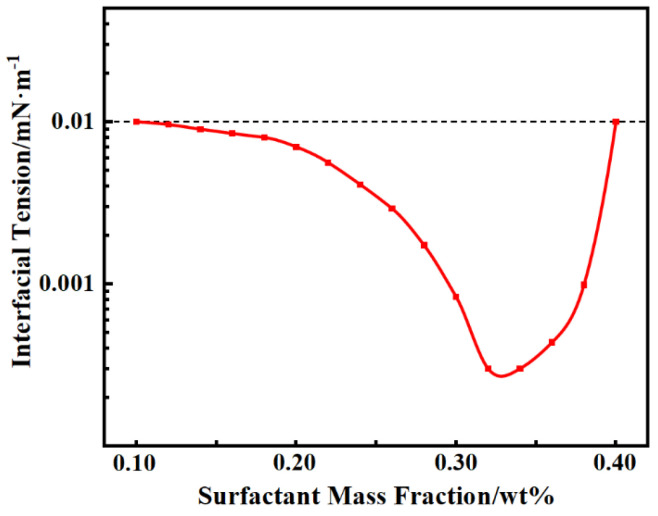
Interface tension curves under different mass fractions of surfactants.

**Figure 5 polymers-16-02004-f005:**
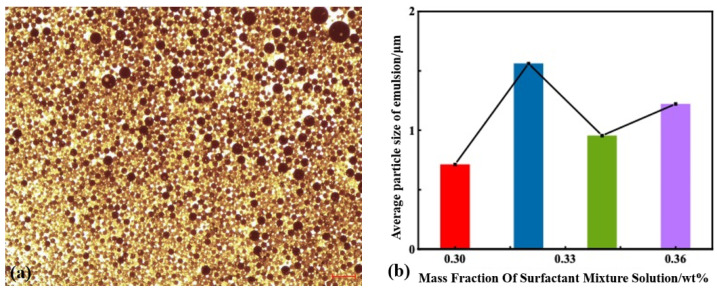
Microscopic image of oil in water system (**a**) and particle size distribution and average particle size of emulsion droplets (**b**).

**Figure 6 polymers-16-02004-f006:**
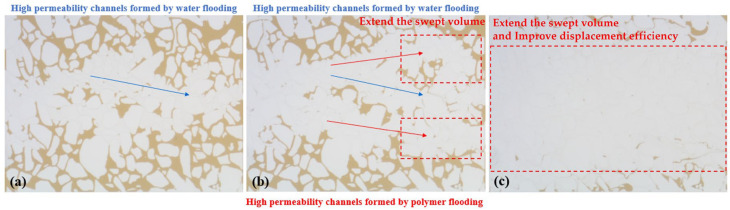
High-permeability channels formed by water flooding (**a**), polymer flooding forms high-permeability channels (**b**) and distribution of remaining oil after SP flooding (**c**).

**Figure 7 polymers-16-02004-f007:**
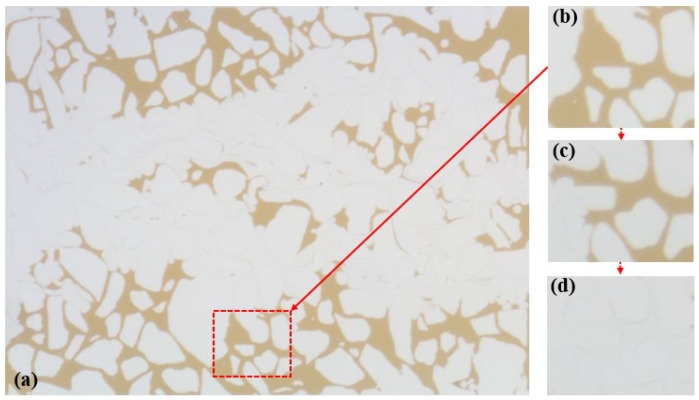
Remaining oil distribution after polymer flooding (**a**), part of the remaining oil (**b**), oil–water flow ratio decrease and driven remaining oil (**c**), completed remaining oil displacement (**d**).

**Figure 8 polymers-16-02004-f008:**
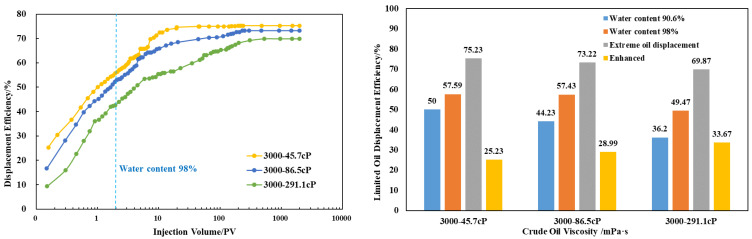
Water flooding experiments of the S Oilfield (3000 mD).

**Figure 9 polymers-16-02004-f009:**
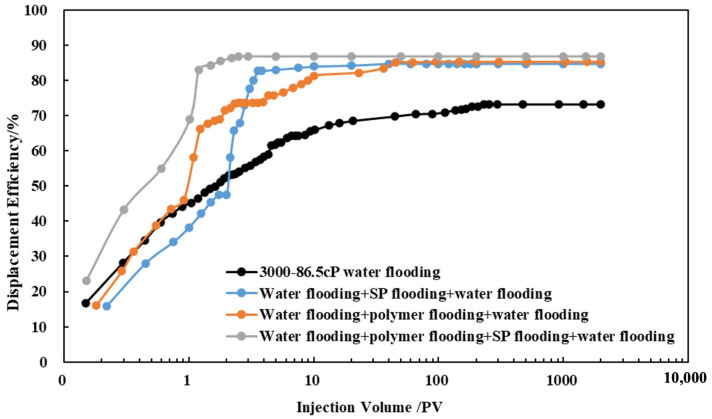
Comparison of displacement experiments (3000 mD).

**Figure 10 polymers-16-02004-f010:**
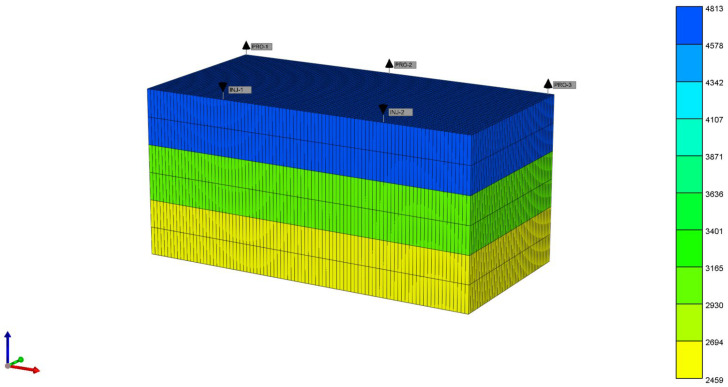
The conceptual model for the high-permeability heterogeneous S Oilfield.

**Figure 11 polymers-16-02004-f011:**
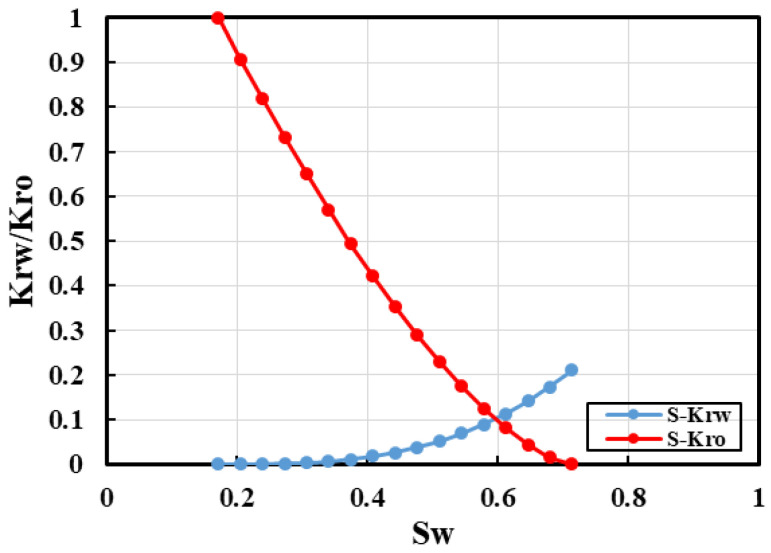
The permeability curve of conceptual model.

**Figure 12 polymers-16-02004-f012:**
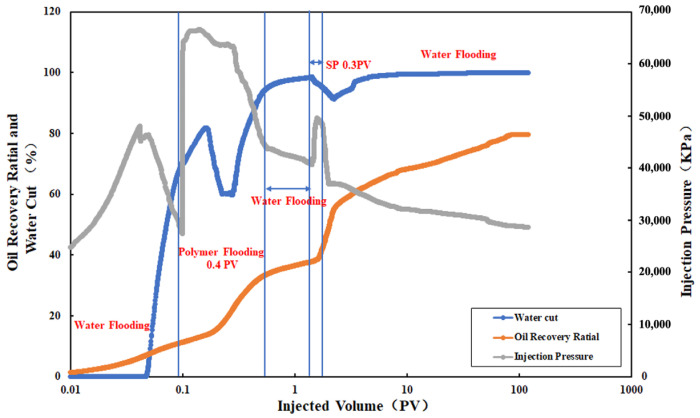
Changes in pressure, recovery rate and water content of SP flooding after polymer flooding.

**Figure 13 polymers-16-02004-f013:**
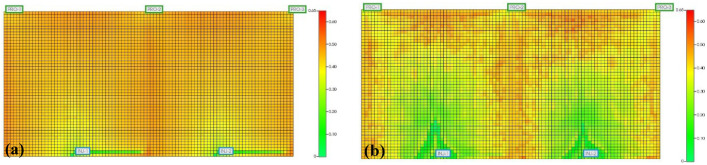
The change in sweep range after 6 months of follow-up of water flooding after polymer flooding (**a**) and 6 months of follow-up of water flooding after SP flooding (**b**).

**Figure 14 polymers-16-02004-f014:**
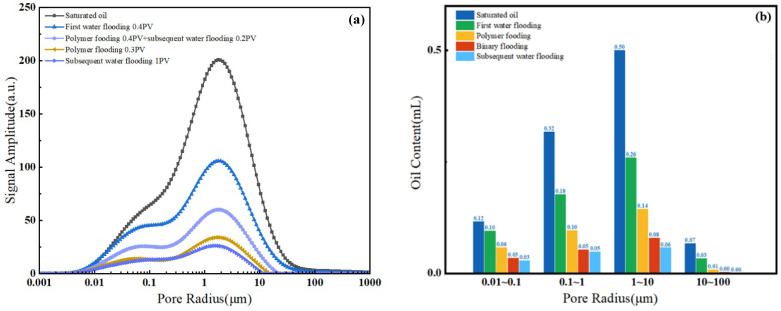
T_2_ spectra for each stage (**a**) and oil content distribution in different pores at different stages (**b**).

**Figure 15 polymers-16-02004-f015:**
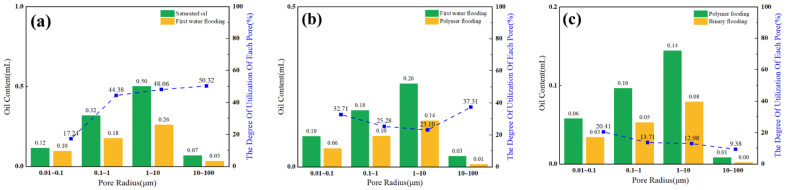
Distribution of oil content and effective use of pore development in water flooding stage (**a**), polymer flooding stage (**b**) and SP flooding stage (**c**).

**Figure 16 polymers-16-02004-f016:**
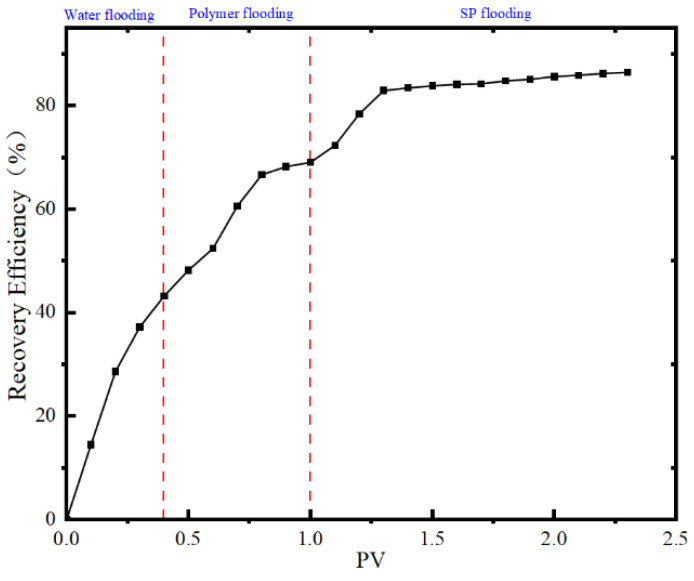
Recovery rates at different stages.

**Figure 17 polymers-16-02004-f017:**
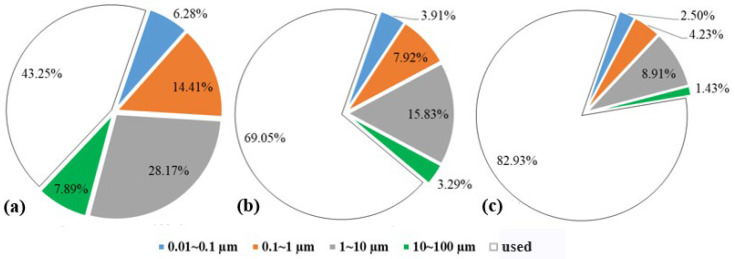
Distribution of remaining oil in water flooding stage (**a**), polymer flooding stage (**b**) and SP flooding stage (**c**).

**Table 1 polymers-16-02004-t001:** Conceptual Model Parameters for the S Oilfield.

Parameter	Taking Values	Parameter	Taking Values
Grid step(m)	500 (direction I); 250 (direction J); 48 (direction K)	Oil saturation	0.65
Number of grids	91 × 50 × 6	Oil viscosity (mPa·s)	86.5
Porosity (%)	32	Stratum water viscosity (mPa·s)	0.5
Permeability variation coefficient	0.35	Polymer injection amount (PV)	0.4
Average permeability(mD)	3419.67	SP injection amount (PV)	0.3

## Data Availability

The original contributions presented in the study are included in the article/Appendix A, further inquiries can be directed to the corresponding author.

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
