# Peer review of "Study on Surfactant–Polymer Flooding after Polymer Flooding in High-Permeability Heterogeneous Offshore Oilfields: A Case Study of Bohai S Oilfield"

_polymers, 2024, doi:10.3390/polym16142004_

Round 1
Reviewer 1 Report
Comments and Suggestions for Authors
The manuscript "polymers-2913972" by Liu et al. reported Study on the Surfactant-polymer Flooding after Polymer Flooding in High-Permeability Heterogeneous Offshore Oilfields: A Case Study of Bohai S Oilfield. After review, the authors have to make minor changes. The authors should refer to the following comments to improve their work:
1. Page 1, line 34: "China, being the largest petroleum producer in chemical EOR projects, primarily relies on polymer flooding." Please cite a reference.
2. Page 2, lines 82-89: "S Oilfield faces several difficulties: Firstly, due to the complex offshore reservoir conditions with multiple vertical control layers in a single well, and limitations imposed by offshore platforms, implementing measures such as layer adjustments and well pattern improvements to address the oil-water issue is challenging. Secondly, the offshore platform lacks freshwater resources required for polymer injection, relying instead on formation water and seawater with elevated salinity. Thirdly, S Oilfield has large well spacing, severe reservoir heterogeneity, and polymer tends to channel into high-permeability layers, complicating the recovery of remaining oil in wells." Please cite a reference.
3. Please explain more about biosurfactant and polymer (In section 2.1). It is written very unintelligibly.
4. What effect can the change of polymer viscosity have on the sweep volume due to temperature and pressure changes?
5. In your opinion, improving the ultimate oil recovery efficiency depends on what factors?
6. Please clearly explain what is the superiority of this manuscript over other studies (in this field). Add this explanation to the manuscript (introduction section).
7. Please show the important factors in enhanced oil recovery in the form of a graphic design.
8. Please replace more up-to-date references.
9. The language of the manuscript should be checked.
Minor editing of English language required.
Author Response
Reviewer #1: The manuscript "polymers-2913972" by Liu et al. reported Study on the Surfactant-polymer Flooding after Polymer Flooding in High-Permeability Heterogeneous Offshore Oilfields: A Case Study of Bohai S Oilfield. After review, the authors have to make minor changes. The authors should refer to the following comments to improve their work.
Response: Thank you for your review to improve the quality of the manuscript. We have revised the manuscript in accordance with the comments and reply one by one.
Comment 1: Page 1, line 34: "China, being the largest petroleum producer in chemical EOR projects, primarily relies on polymer flooding." Please cite a reference.
Response 1: Thank you for your comments. We have added references to this sentence (line 37).
Comment 2: Page 2, lines 82-89: "S Oilfield faces several difficulties: Firstly, due to the complex offshore reservoir conditions with multiple vertical control layers in a single well, and limitations imposed by offshore platforms, implementing measures such as layer adjustments and well pattern improvements to address the oil-water issue is challenging. Secondly, the offshore platform lacks freshwater resources required for polymer injection, relying instead on formation water and seawater with elevated salinity. Thirdly, S Oilfield has large well spacing, severe reservoir heterogeneity, and polymer tends to channel into high-permeability layers, complicating the recovery of remaining oil in wells." Please cite a reference.
Response 2: Thank you for your comment. We have added references to this paragraph (lines 94-101).
Comment 3: Please explain more about biosurfactant and polymer (In section 2.1). It is written very unintelligibly.
Response 3: Thank you for your significant suggestions. We have improved the expression in the revised manuscript. (line 121-127 and 128-133)
Comment 4: What effect can the change of polymer viscosity have on the sweep volume due to temperature and pressure changes?
Response 4: Thank you for your comment. The viscosity of the polymer decreases with the increase of temperature and increases with the increase of pressure. The increase of viscosity can improve the mobility ratio and enlarge the sweep volume of the polymer. Thus, the viscosity of the surfactant-polymer system under the conditions of temperature and pressure of S Oilfield was given in our manuscript (lines 121-127).
Comment 5: In your opinion, improving the ultimate oil recovery efficiency depends on what factors?
Response 5: Thank you for your comment. The effect of the enhance oil recovery decided by the sweep volume and oil displacement efficiency of the injection medium. Both the expansion of swept volume and the improvement of the displacement efficiency can improve the final oil recovery. In conventional oil-wet reservoirs, reducing interfacial tension and changing wettability are conducive to improving oil displacement efficiency. The SP flooding can simultaneously expand the swept volume by increasing the viscosity of the displacement phase, improve the oil displacement efficiency by reducing the interfacial tension, and synergistically improve the oil recovery rate. Therefore, the mechanism of expanding sweep volume and improving oil displacement efficiency of SP flooding was firstly analyzed in this manuscript. Then the enhanced oil recovery potential of SP flooding in Bohai S oilfield was discussed through core displacement and numerical simulation. (3. Results and Discussion)
Comment 6: Please clearly explain what is the superiority of this manuscript over other studies (in this field). Add this explanation to the manuscript (introduction section).
Response 6: Thank you for your significant suggestions. We have added an explanation of the superiority of this manuscript in the introduction section. (lines 115-118)
Comment 7: Please show the important factors in enhanced oil recovery in the form of a graphic design.
Response 7: Thank you for your significant suggestions. We have modified the Fig.6 and show the important factors in enhanced oil recovery in our manuscript.
Comment 8: Please replace more up-to-date references.
Response 8: Thank you for your significant suggestions. We have added five new references in our manuscript. (Reference)
Comment 9: The language of the manuscript should be checked.
Response 9: Thank you for your significant suggestions. We have revised issues and improved the grammar in our manuscript.
Reviewer 2 Report
Comments and Suggestions for Authors
Dear Authors! I reviewed your manuscript "Study on the EOR of SP Flooding after Polymer Flooding in High-Permeability Heterogeneous Offshore Oilfields: A Case Study of Bohai S Oilfield" and found that it consitutes a good piece of scientifically sound work in Oil Recovery Science and practice. I have only few remarks - mainly technical given below. On the other hand, I have strong doubts that your good manuscript does fit the agenda of MDPI Polymers journal and especially Section Polymer Processing and Engineering, Special Issue New Studies of Polymer Surfaces and Interfaces. I hope Editorial Board finds right decision.
Remarks:
Abbrevations in abstract are not disclosed.
Lines
69. taking -> taken
73. northeast -> north-east
222. water-in-oil (W/O) term contradicts to the term in Caption for Figure 5, where oil-in-water is given. I would agree with the term oil-in-water, since water cut of 90 % leaves no doubts.
Figure 5 b. - No statistical dispersion/standard deviation for average droplet sizes is given - this undermines any interpreting. Also speculations on droplet size distribution curves when no single curve is given are too weak (lines 230-237).
281. "was around 230 pore volumes (PV)" - in the Figure 8a one can see PV >3000. I guess there is a bad phrasing to be improved.
336-400. Good piece of work having no relations with Polymers. Do you really need it in the manuscript?
Author Response
Reviewer #2: Dear Authors! I reviewed your manuscript "Study on the EOR of SP Flooding after Polymer Flooding in High-Permeability Heterogeneous Offshore Oilfields: A Case Study of Bohai S Oilfield" and found that it consitutes a good piece of scientifically sound work in Oil Recovery Science and practice. I have only few remarks - mainly technical given below. On the other hand, I have strong doubts that your good manuscript does fit the agenda of MDPI Polymers journal and especially Section Polymer Processing and Engineering, Special Issue New Studies of Polymer Surfaces and Interfaces. I hope Editorial Board finds right decision.
Response: Thank you for your significant suggestions. We revised the full text according to your suggestion and improved the quality of the manuscript.
Comment 1: Abbrevations in abstract are not disclosed.
Response 1: Thank you for your comments. We have modified the abstract by replacing “SP” with “surfactant-polymer” and “EOR” with “enhance oil recovery”. (lines 9-26)
Comment 2: Line 69. taking -> taken and, Lines 73. northeast -> north-east
Response 2: Thank you for your comment. We have changed “taking” to “taken” in line 79, and “northeast” to “north-east” in line 83.
Comment 3: Line 222. water-in-oil (W/O) term contradicts to the term in Caption for Figure 5, where oil-in-water is given. I would agree with the term oil-in-water, since water cut of 90 % leaves no doubts.
Response 3: Thank you for your significant suggestions. We have modified the “water-in-oil” to “oil-in-water” in our manuscript. (line 246)
Comment 4: Figure 5 b. - No statistical dispersion/standard deviation for average droplet sizes is given - this undermines any interpreting. Also speculations on droplet size distribution curves when no single curve is given are too weak (lines 230-237).
Response 4: Thank you for your significant suggestions. Due to negligence, we are unable to provide the corresponding experimental results. The statistical deviation has been supplemented in our manuscript, and the cumulative curve of particle size distribution was provided.
Comment 5: Line 281. "was around 230 pore volumes (PV)" - in the Figure 8a one can see PV >3000. I guess there is a bad phrasing to be improved.
Response 5: Thank you for your comment. The 230 pore volume (PV) is the point at which all viscous crude oil was displaced out, and the water cut reached 100% after 230 PV.
Comment 6: Line 336-400. Good piece of work having no relations with Polymers. Do you really need it in the manuscript?
Response 6: Thank you for your significant suggestions. The main purpose of this part is to prove through experiments that SP flooding can reduce the injected PV when the ultimate oil displacement efficiency is achieved. As a result, the oil displacement efficiency can be improved and the cost of oil field development can be reduced. This result shows that SP flooding has great advantages in many aspects compared with water flooding. We do believe that this part of the content is meaningful to our manuscript.
Reviewer 3 Report
Comments and Suggestions for Authors
The paper topic is interesting. I have the following comments to improve the quality of the manuscript:
· What do you mean by “after polymer flooding”? Does it mean that the water cut reaches back to the original condition before polymer flooding? I recommend specifying the definition in the introduction section.
· I suggest showing the current status of the field after polymer flooding. Data for residual oil and water cut.
· Introduction section should be enhanced. Is there any report of similar application worldwide?
· In this work, a synthetic micromodel was used, so the type of the rock and operational conditions (like pressure) is different from the field under the study. So why do authors believe that the results of this work can be used for a specific field? A clear discussion is required to be added to the paper to show the logic of the selected methodology to answer the problem statement.
· Write in Figure 7 caption description of each figure (a, b, ,c, d)
· Moving back to the methodology, what is the reason of selecting 3 different oils for the flooding experiments. Are they all reprehensive of the oil in the field? I believe that the title of the paper should be revised. This manuscript is not directly related to the specific mentioned field and the field conditions were not considered in the designed tests.
· Revise Figure 8, what is the meaning of amplification? And limit oil displacement? Please use the correct terminology accepted in petroleum engineering literature. The results of waterflooding is a bit unusual, when is the breakthrough time? After 10 PV of injection? How is it possible?
· On Figure 9, specify the time that an EOR method is started. Also what is the difference between blue and orange curves. Generally results should be presented in a better and more clear way. From what presented in this figure, the benefit of P/SP flooding is not clear.
· Is the result presented on Figure 16 on the similar core with other tests? What is your observation compare to results of Figure 9?
Comments on the Quality of English Language
There are some words in the manuscript that are not among acceptable technical words in petroleum engineering and it looks like wrong translation.
Author Response
Reviewer #3: The paper topic is interesting. I have the following comments to improve the quality of the manuscript:
Response: Thank you for your review to improve the quality of the manuscript. We have revised the manuscript in accordance with the comments and reply one by one.
Comment 1: What do you mean by “after polymer flooding”? Does it mean that the water cut reaches back to the original condition before polymer flooding? I recommend specifying the definition in the introduction section.
Response 1: Thank you for your comments. Sorry for the misunderstanding caused by our inaccurate statement. After polymer flooding refers to the late stage of polymer flooding with a high water cut, and polymer flooding was difficult to effectively displace crude oil during this stage. Therefore, it is considered to further improve the recovery degree of reservoir by SP flooding. We have been explained this phrase in our manuscript. (lines 62-63)
Comment 2: I suggest showing the current status of the field after polymer flooding. Data for residual oil and water cut.
Response 2: Thank you for your comment. We have supplemented the current development status of S oilfield in our manuscript, including water cut, current oil recovery, etc. (lines 86-88)
Comment 3: Introduction section should be enhanced. Is there any report of similar application worldwide?
Response 3: Thank you for your significant suggestions. We have supplemented the description of similar studies in the introduction. The investigated literature has studied the exploitation technology and difficulties of offshore oilfields. Shuang Liang in order to improve the development effect of offshore oil field at high water cut stage, studied the technology of nitrogen foam flooding and polymer microsphere flooding [23]. Lizhen Ge clarified the mechanism of weak gel flooding and remaining oil distribution in LD oilfield [24]. There are relatively few studies on the application of SP flooding technology in offshore oilfields. (lines71-77)
Comment 4: In this work, a synthetic micromodel was used, so the type of the rock and operational conditions (like pressure) is different from the field under the study. So why do authors believe that the results of this work can be used for a specific field? A clear discussion is required to be added to the paper to show the logic of the selected methodology to answer the problem statement.
Response 4: Thank you for your significant suggestions. The microscopic model is an experiment carried out under the temperature and pressure conditions consistent with the actual reservoir. At the same time, the shape of the pores is also consistent with the actual core. Thus, the experimental results have good similarity with the actual reservoir. We have supplemented the specific experimental conditions in our manuscript. (line 180-183)
Comment 5: Write in Figure 7 caption description of each figure (a, b, ,c, d)
Response 5: Thank you for your comments. We have modified the Figure 7 caption description of each figure (a, b, c, d)
Comment 6: Moving back to the methodology, what is the reason of selecting 3 different oils for the flooding experiments. Are they all reprehensive of the oil in the field? I believe that the title of the paper should be revised. This manuscript is not directly related to the specific mentioned field and the field conditions were not considered in the designed tests.
Response 6: Thank you for your comments. There are blocks with different viscosities in this block of the target oilfield. Therefore, we discussed the displacement experiments of crude oil with different viscosities to better study the effect of viscosity on the binary flooding effect in actual reservoirs and provide better guidance for actual reservoir development. Specific reservoir conditions have been supplemented in the manuscript. (lines 86-88)
Comment 7: Revise Figure 8, what is the meaning of amplification? And limit oil displacement? Please use the correct terminology accepted in petroleum engineering literature. The results of waterflooding is a bit unusual, when is the breakthrough time? After 10 PV of injection? How is it possible?
Response 7: Thank you for your comments. We have modified the inappropriate words and replaced them with petroleum engineering language. The “amplification” is replaced by “enhanced”, and the “limit oil displacement” is replaced by “ultimate oil recovery efficiency”. The breakthrough time reached around 2PV, with a water cut of 98 %, but the water cut reached 100% after 10 PV. Because the experimental core is high permeability core, and the viscosity of crude oil is not particularly high, the crude oil is relatively easy to be displaced by water. Therefore, the injection volume of water flooding from the 98% water cut to 100% water cut was only 10PV. The injection volume of this process also increased significantly with the increase of crude oil viscosity. When the crude oil viscosity was 86mpa·s, the injection volume of this process was about 100PV. When the crude oil viscosity was 290mpa·s, the injection volume of this process was about 200PV.
Comment 8: On Figure 9, specify the time that an EOR method is started. Also what is the difference between blue and orange curves. Generally results should be presented in a better and more clear way. From what presented in this figure, the benefit of P/SP flooding is not clear.
Response 8: Thank you for your comments. The experiment simulated the development process of the oilfield. Therefore, the EOR measure begins to be implemented at about 70% of water cut. Polymer flooding started at 0.88PV, and SP flooding started at 1.08PV. In Fig. 9, the blue curve represents SP flooding, and the orange curve represents polymer flooding. From Fig. 9, we can find that although the ultimate displacement efficiency (100% water cut) of the polymer and SP flooding are similar, the injected PV is significantly reduced for the SP flooding. When the water cut achieve 100%, the injected PV of SP flooding was only 2 PV, while the polymer flooding was 20 PV. This is due to the fact that the surfactant improves the sweep efficiency by decreasing the interfacial tension between oil and water. Secondly, when water cut is 98%, the oil displacement efficiency of polymer flooding is 73.54%, but the oil displacement efficiency of SP flooding is 82.79%. These results proved the benefit of SP flooding.
Comment 9: Is the result presented on Figure 16 on the similar core with other tests? What is your observation compare to results of Figure 9?
Response 9: Thank you for your comments. Fig. 9 is the result of conventional oil displacement experiments. In Fig. 16, the cores that used to conduct nuclear magnetic displacement experiment were similar to conventional oil displacement experiments. Since the two groups of experimental materials and processes are basically the same, the trend of experimental results we observed were similar.
Round 2
Reviewer 3 Report
Comments and Suggestions for Authors
Comments are addressed.